# Trust in Institutions, Not in Political Leaders, Determines Compliance in COVID-19 Prevention Measures within Societies across the Globe

**DOI:** 10.3390/bs12060170

**Published:** 2022-05-30

**Authors:** Ryan P. Badman, Ace X. Wang, Martin Skrodzki, Heng-Chin Cho, David Aguilar-Lleyda, Naoko Shiono, Seng Bum Michael Yoo, Yen-Sheng Chiang, Rei Akaishi

**Affiliations:** 1Center for Brain Science, RIKEN, Wako 351-0198, Japan; david.aguilarlleyda@riken.jp (D.A.-L.); naoko.shiono@riken.jp (N.S.); rei.akaishi@riken.jp (R.A.); 2Economics Department, State University of New York at Binghamton, Binghamton, NY 13902, USA; xwang222@binghamton.edu; 3RIKEN Interdisciplinary Theoretical and Mathematical Sciences Program, RIKEN, Wako 351-0198, Japan; m.skrodzki@tudelft.nl; 4Computer Graphics and Visualization, Department of InSy/EEMCS, Delft University of Technology, P.O. Box 5031, 2600 GA Delft, The Netherlands; 5Institute of Sociology, Academia Sinica, Taipei 11529, Taiwan; hengchin.cho@gmail.com (H.-C.C.); chiangys@gate.sinica.edu.tw (Y.-S.C.); 6Center for Neuroscience Imaging Research, Institute for Basic Science (IBS), Suwon 16419, Korea; sbyoo.ur.bcs@gmail.com; 7Department of Biomedical Engineering, Sungkyunkwan University (SKKU), Suwon 16419, Korea

**Keywords:** COVID-19, institutional trust, social trust, public health compliance, transparency, political trust

## Abstract

A core assumption often heard in public health discourse is that increasing trust in national political leaders is essential for securing public health compliance during crises such as the COVID-19 pandemic (2019–ongoing). However, studies of national government trust are typically too coarse-grained to differentiate between trust in institutions versus more interpersonal trust in political leaders. Here, we present multiscale trust measurements for twelve countries and territories across the West, Oceania and East Asia. These trust results were used to identify which specific domains of government and social trust were most crucial for securing public health compliance (frequency of mask wearing and social distancing) and understanding the reasons for following health measures (belief in effectiveness of public health measures). Through the use of linear regression and structural equation modeling, our cross-cultural survey-based analysis (*N* = 3369 subjects) revealed that higher trust in national and local public health institutions was a universally consistent predictor of public health compliance, while trust in national political leaders was not predictive of compliance across cultures and geographical regions. Institutional trust was mediated by multiple types of transparency, including providing rationale, securing public feedback, and honestly expressing uncertainty. These results highlight the importance of distinguishing between components of government trust, to better understand which entities the public gives the most attention to during crises.

## 1. Introduction

Trust can be a powerful predictor of human behavior in modern societies, as trust is the glue that maintains long-term and beneficial social relationships between both individuals and groups, despite the inherent risk that presents in relying on others [1,2]. However, trust is notoriously hard to define, despite broad public and scientific consensus in the conceptual importance of trust for fostering social cohesion and cooperation [1,2,3,4,5]. Even if researchers reached consensus on a narrow definition of trust (unlikely as this is, [6]), such a narrow definition likely would disagree with the multitude of ways trust is used and interpreted within the public vernacular [7]. Thus, overly narrow academic definitions of complex social phenomenon may actually hinder ecological scientific investigations such as survey-based research. However, across sub-fields of trust research, commonalities in what attributes that trustworthy individuals, or groups (e.g., institutions), encompass include the subject of one’s trust possessing competence, benevolence, predictability, and/or integrity [6]. The process by which such trustworthiness is evaluated (i.e., the process of deciding whether to trust a person or institution) can include analysis of personality and disposition, structural features of accountability, affect and attitude, expectations and compliance with accepted social norms, intentions and motivations (and conflicts of interest), and/or recent history of specific behaviors [2,6,8,9]. 

Additionally, public trust in individuals or institutions can be readily and consistently assessed by researchers through self-report survey questionnaires, allowing measurements of trust to be correlated with a variety of behaviors and outcomes [10,11,12]. Researchers have used cross-cultural variations in different categories of trust to explain differences in economic outcomes, social norms, social capital, public health outcomes, etc. across communities and nations [2,8,9,11,13,14]. Cooperative efforts in large societies are fundamentally built on a complex multi-tiered structure of trust and cooperation, with separate spheres including local community trust, trust of experts and specialists, trust of political leaders (elected officials in democracies), and trust in the (unelected) bureaucracies and agencies that actually run the daily life of a society [9,15,16]. Such large-scale cooperative efforts within societies are especially critical for coordinating disaster responses to deal with crises (e.g., pandemics, wars, natural disasters) [9,11,13]. 

A common, but potentially misleading [16], implicit assumption frequently seen in popular discourse and trust research across disciplines is that increasing trust in national political leaders is critical for a nation’s success broadly [17,18,19,20,21,22,23]. Overstating the influence of national political leaders on national outcomes generally, whether the outcome is good or bad, is an ever-present risk in analysis however, due to a core but dangerous cognitive bias in humans towards simple explanations. Particularly, humans are biased towards simple latent scope explanations where one cause or agent (e.g., heads of state) can be assigned as the primary explanation for many problems at once [24]. In reality, at the national scale, policy successes or failures often cannot be explained by the type of simple explanations we become accustomed to hearing in typical media coverage [15,25]. Furthermore, particularly in democratic societies, the quality of national political leaders themselves is more likely a reflection of the general state of institutions or the broader society [8,26,27]. In fact, the foundation of democratic systems is the recognition that the inherent conflicts of interests involved with holding political office necessitate strong institutional checks and balances to maintain a well-functioning government [15]. Thus, some baseline interpersonal-level distrust of elected political leaders is healthy and actually a requisite for a functional democracy [28]. The steady drop in government trust during the past few decades in many developed democracies has been bemoaned as a dangerous trend [8]. In fact, as pointed out by several trust researchers, this trend may just be a sign of a more informed and educated citizenry adopting healthier skepticism for their political leaders [19]. 

In major crises that require disaster management, it is actually scientifically oriented government agencies that are most critical for using the current scientific consensus and their institutional experience to assist the public (e.g., public health and disease control, environmental management, food and drug safety, etc.) [16,29,30,31]. Specifically, public health officials and agencies are consistently among the most trusted parts of governments [10,30,32]. On the one hand, citizens want their political leaders to interpersonally resonate with them, and to consider both the personal values and interests of the public as well as expert advice during complex decision making. On the other hand, what the public wants from their scientifically oriented government agencies can be quite different: transparent, non-ideological communications and instructions, directly from scientific experts, on complicated and evolving crises [33,34,35]. Therefore, the ultimate responsibility for crisis management actually resides within the national and local agencies, and their historical competence and consistency especially during crises is what fosters institutional trust [32,33,34]. 

Thus, different facets of government trust (e.g., institutions versus political leaders) may be differentially influenced by separate contributions from the core dimensions of (1) *interpersonal trust* at the individual-level towards politicians, agency heads or staff, etc., (2) *institutional trust* related to the assigned societal roles, competency and histories of institutions, and (3) *political trust* related to one’s preferred political ideology and political party [28,36,37,38]. National political leader trust is more influenced by the interpersonal and political dimensions, while trust in public health agencies is more comprised of the institutional trust dimension, a dimension that may actually be strengthened when it is *not* politicized [16,39]. Additionally, transparency has been found to have sometimes mixed [35,40,41,42], but usually positive effects on both institutional trust and political leader trust in public health crises [10,12,43,44,45]. Given these considerations, we build our study on the concept that trust towards agencies and trust towards political leaders are fundamentally different types of trust, and possibly affected by transparency in different ways. Furthermore, institutional trust over political leader trust should be the stronger predictor of public behavior during public health crises. Therefore, these different types of trust in government must be examined separately [10,15,16,28]. 

Given that particular crises rarely generalize well across geographical regions, studying these separate government trust components during crisis response, in a controlled cross-cultural way, has been a historical challenge until now. The recent COVID-19 coronavirus pandemic (2019–ongoing) may be a unique, and possibly the first, time in human history (after the Spanish flu), that all nations have been faced with the same public health challenge around the world [19]. Thus, the tragic ongoing pandemic has globally created unusually controlled, cross-cultural study conditions for exploring the effects of trust on public health behaviors across the world. 

To better understand the effects of different components of trust on public behavior during a crisis, in this work we have developed a fine-grained survey to probe multiple scales of trust in society during the COVID-19 pandemic. We measure both social (e.g., family, friends, strangers) and government-related (e.g., institutions and political leaders) trust categories. We surveyed a set of countries and territories across the West, East Asia and Oceania: the United States of America, the United Kingdom, Canada, Germany, Spain, Israel, Australia, New Zealand, Japan, South Korea, Taiwan, and China (12 countries/territories, *N* = 3369 subjects). These twelve countries/territories were chosen based on both cultural diversity and performance variation within two geographic regions that are often compared as contrasting cultural blocs in sociological and cultural psychology research (East Asia and “the West”) [2,9]. In East Asia: Taiwan, Japan, South Korea, and China generally had much lower per capita death rates than “Western” countries, with the United States and United Kingdom being among the worst in the world for recorded per capita death rates (as of our late 2020 data collection period) [46,47]. Prior leading cultural psychology work had explained these trends with simple Asian “collectivism” versus Western “individualism” stories [48,49], but clear exceptions existed such as New Zealand within the Western countries which had almost zero COVID-19 deaths despite a highly individualistic culture [45]. Generally, within Western countries, the better-performing Australia and New Zealand had the advantage of being isolated island nations that were easier to manage in a pandemic context than the generally worse-performing Europe, North America and Israel, a factor which may partially explain differences in pandemic performance [50]. Additionally, even within collectivist cultures, substantial variation existed in institutional strategy in ways that seem unrelated to collectivism (e.g., China had a top-down enforced centralized zero-COVID strategy, while Japan had a decentralized mitigation strategy that relied more on social norms for enforcement) [29,51]. Furthermore, East Asian cultures importantly had substantial institutional experience with prior coronavirus pandemics during the mid-2000s SARS coronavirus outbreak and the mid-2010s MERS coronavirus outbreak [52,53,54,55,56]. These recent pre-COVID coronavirus-based respiratory disease experiences in East Asia likely helped improve the preparedness of, and public confidence in, public health institutions in this region, especially for future coronavirus outbreaks including the COVID-19 pandemic. For example, Taiwan’s COVID-19 response was largely guided by detailed institutional response plans developed during the earlier SARS outbreak [29]. Thus, generally, we wished to probe deeper into more nuanced trust networks as a further sociocultural explanation beyond collectivism and individualism.

We hypothesized (H1) that trust in public health institutions, which are staffed by experts, will be one of the most consistent and strong predictors of public health measure compliance and beliefs: (1) mask wearing compliance, (2) social distancing compliance, and (3) beliefs in the effectiveness of these public health measures. We chose these compliance variables to test which trust factors induce compliance in the most important infection prevention behaviors [57]. Furthermore, we examined which trust factors make people personally believe those measures are effective, to further probe whether subjects were complying with public health measures due to being personally convinced the measures were effective versus other reasons (e.g., due to government enforcement rather than personal belief). Interpersonal trust in national political leaders was expected to be a weak predictor of these compliance behaviors, as substantial medical expertise that leaders typically lack was required to understand and manage the complex COVID-19 crisis [28]. However, in-depth institutional case studies for each country/territory are beyond the scope of the current work, as strategies and performance varied substantially within countries/territories, and often changed drastically across short time frames, during the COVID-19 pandemic [23,29,51,58,59,60,61,62,63,64]. Establishing causal links between specific policy decisions and self-report trust measures from a survey is not a trivial undertaking and surpasses the limits of our data, though is an important direction for future research to pursue. Additionally, it is important for future work to investigate trust networks in other understudied geographic regions within the COVID-19 context, such as South America and Africa [65,66,67], and we hope this current work can provide guidance in such broader studies. Thus, overall, in this work we sought to explore the centrality of different types of government trust within each country/territory in securing compliance in preventative public health behavior, using a more generalized sociological view that we hope can help guide more applied case study approaches. 

The specific trust categories included in the survey are trust in national political leaders, institutional trust in both national and local public health institutions, local community trust, trust in strangers, social media trust, traditional news media trust, trust in employers, trust in science, and trust in the World Health Organization—all framed in the context of the COVID-19 pandemic (Appendix A). We also included more traditional general trust measures to compare to previous results [68,69]. As control measures, we included demographic information. Last, to test the mediating effects of transparency-related policy on local and national institutional trust, and trust in national political leaders, in order to relate transparency to public compliance with preventative measures (H2), we examined survey responses about the degree to which governments practiced several categories of institutional transparency during the COVID-19 pandemic. These transparency measures included topics such as providing rationale, securing public feedback, and designing policy for diverse contexts in society (Appendix A). Transparency is generally a multifaceted concept that cannot be simply reduced to one narrow policy approach [70]; thus, we have focused primarily on transparency in decision making and policy contents. We treated transparency as a broader latent variable which underlies the separate transparency components measured in our survey responses.

In our results we found that higher levels of both local and national institutional trust reliably predicted stronger compliance in public health measures, as well as stronger beliefs in the effectiveness of those measures, across the cultures studied, while trust in national political leaders does not. Transparency as a policy only improved public health behaviors and beliefs when transparency improved government trust, especially national institutional trust. Higher government transparency uncoupled to trust was found to be neutral or even harmful if the public did not trust the government. Our findings suggest that it is critical for nations to cultivate and maintain trust between the public, government institutions, and experts in order to elicit better society-level cooperation from the public, especially during major crises. Moving forward, researchers should seek to reconcile broader international sociological results of trust, culture, etc. (such as the ones presented in this work), with more fine-grained and time-dependent case studies of institutional performance and knowledge strategies within individual countries/territories [71], as both lines of research capture important information but these lines of research are rarely combined within a single study.

## 2. Materials and Methods

### 2.1. Survey Design and Data Collection

Our survey was designed to examine how government and social trust measures interact with government transparency to predict public health compliance behaviors. Survey data were taken from twelve countries/territories in each of their official national languages, between October 2020 and December 2020. Prolific was used to collect data from Australia (*N* = 253), Canada (*N* = 254), Germany (*N* = 289), Israel (*N* = 234), New Zealand (*N* = 207), Spain (*N* = 253), the United Kingdom (*N* = 273), and the United States (*N* = 344). Crowdworks was used to collect data from Japan (*N* = 318). QuestionPro was used to collect data from South Korea (*N* = 299) and China (*N* = 406). Taiwanese data were collected independently by Y.-S.C. using representative snowball sampling (*N* = 239). Full survey data and the translated surveys for each country and territory are included in Appendix A. Answering all survey questions was mandatory, so there were no missing question responses. 

To prepare the main survey trust variables for analyses, we grouped them into the following categories based both on statistical and conceptual correlations: national institutional trust in public health institutions (Q73 and Q74, *r =* 0.77), national political leader trust (Q19 and Q72, *r =* 0.72), local institutional trust (Q55 and Q57, *r* = 0.48), local community trust (Q61 and Q62, *r* = 0.53), trust in strangers (Q58 and Q63, *r* = 0.45), trust in employers (Q64), social media trust (Q56), traditional news media trust (Q59), trust in science (Q97), and trust towards the World Health Organization (Q93) (*r* is the Pearson correlation coefficient, “QX” notation refers to the original survey question number in Appendix A, with the main questions used in analysis also summarized in the Appendix A). Here, “national” means the highest level of governance within a country/territory. Correlation matrices between all variables we analyzed are summarized in Appendix A. 

### 2.2. Linear Regression

Our primary research objective was identifying which component(s) of government trust comprised a more consistent and stronger predictor of public compliance in preventative measures as measured by mask wearing compliance, social distancing compliance, and beliefs in the effectiveness of these public health measures. For this quantitative analysis, we first used the ordinary least-squares (OLS) method to study the association between the trust variables and respondents’ compliance with health prevention measures. More specifically, we estimate the following equation:$$y_{i}=\alpha_{0}+\beta \cdot Trust_{i}+\gamma \cdot X_{i}+\delta \cdot Z_{i}+country+\epsilon_{i}$$
where $y_{i}$ is the outcome variable of our primary interest, namely the compliance with mask-wearing, social distancing and the perceived effectiveness of public measures. $Trust_{i}$ is a vector of applied trust measures. $X_{i}$ is a vector of control variables which contains basic demographic information. $Z_{i}$ is a vector that contains the more abstract measures of general trust and national identity, and *i* is the individual subject index. A set of dummy variables were also included to capture the country/territory-specific trend.

During the regression, 27 subjects (approximately 1–5 per country/territory) were further excluded based on writing “other” or “prefer not to say” for gender, as gender was found to have a significant effect in the regression. Generally, studying non-traditional gender effects is difficult at the global scale in surveys due to language and cultural differences. Additionally, in the regression, 14 additional subjects were omitted from Taiwan for failing to provide income information in Q8 (a “prefer not to answer” was accidentally provided as a possible response for income just in Taiwan). 

Heterogenous per-country/territory results were calculated by marginal effect analysis in a second regression that included interaction terms between each trust variable and a country/territory factor (e.g., “UK” = 1 if the subject is from the UK and 0 else) (Appendix A). The United States’ regressors were chosen as the baseline country for marginal effects comparison as it was the democracy with the largest number of subjects. 

### 2.3. Structural Equation Modeling

Our secondary research objective was to test how transparency influences public health compliance by comparing the direct effects of transparency on compliance, with the indirect effects of transparency on compliance through transparency’s influence on each type of government trust. We fit a set of survey variables to three SEM models of one government trust-related latent variable each, specified as follows: (1) national institutional trust from the indicator variables Q73 and Q74, (2) national political leader trust from Q19 and Q72, and (3) local institutional trust from Q55 and Q57. In all models, six variables (Q80, Q81, Q82, Q83, Q84, Q85) measure the latent variable ‘transparency’, and three variables (Q32, Q34, Q42–Q43) measure our dependent variable ‘public health behavior’. 

We fit a causal relation from transparency to each type of government trust, then to public health behaviors. In other words, transparency is the exogenous variable, government trust is the mediator variable, and public health behavior is the dependent variable (indirect effects model). Additionally, direct effects were tested by an additional causal relation directly from transparency to public health behavior. Standardized coefficients are reported per pathway, with Root-Mean-Square Error of Approximation (RMSEA) and the Comparative Fit Index (CFI) used as the primary measures of model fit quality. RMSEA and CFI are each among the most common quality-of-fit metrics used to evaluate structural equation models, as RMSEA has the advantage of allowing confidence intervals around its value to be calculated and CFI is robust across sample sizes [72,73,74]. RMSEA values are evaluated according to the following ranges: close fit (0.00–0.05), fair fit (0.05–0.08), mediocre fit (0.08–0.1) and poor fit (over 0.10), while CFI > 0.90 is an acceptable fit and CFI > 0.95 is an excellent fit [72,73]. 

## 3. Results

For an outline of the results section, our primary research objective was testing which component of government trust was a more consistent and powerful predictor of public compliance in preventative measures as measured by mask wearing compliance, social distancing compliance, and beliefs in the effectiveness of these public health measures. The two components of government trust studied were trust in public health institutions and trust in political leaders. We hypothesized first (H1) that trust in public health institutions would be the dominant predictor and second (H2) that transparency is a powerful modulator of government trust in a public health context. We tested the first hypothesis (H1) through regression analysis and the second hypothesis (H2) through structural equation modeling. These hypotheses are not alternative hypotheses but rather the components of the larger idea in our study. Regression coefficient magnitude, sign, statistical significance and marginal effects of each type of government trust were used to test the first hypothesis (H1), including by identifying which type of government trust had more consistently strong positive coefficients for predicting public health measure compliance. Then, for H2, structural equation modeling probed this hypothesis by examining which component of government trust was most strengthened by transparency for the outcome of improving public health compliance. Signs, magnitudes and significance of relational links in the structural equation modeling pathways were used to test the hypothesis, specifically by comparing the indirect pathway of transparency, trust, and compliance with the direct pathway of transparency to compliance. These main analyses were presented in Section 3.1. Primary Results. Additionally, the secondary trust and demographic questions used in the regression analysis allowed us to replicate and comment on several major trust-related COVID-19 findings from prior work, thus helping to solidify our baseline survey data quality. These additional brief analyses are provided in Section 3.2. Secondary Results. 

### 3.1. Primary Results

Individual public health decisions result from multiscale trust considerations within a broader social and institutional hierarchy in a society (Figure 1). To identify which types of interactions were most important in securing public health compliance, we first ran an ordinary least-squares (OLS) linear regression model (Table 1). This regression model included social, institutional, and informational trust (Appendix A), and control demographics measures (gender, age, education, income, medical-related work or education experience, political preference, religiosity, number of children, financial difficulty during the COVID-19 pandemic) (Table 1). The first two public health behavior variables were measured in percent of time that one wore masks and social distanced properly when in a situation that health experts would advise to take each prevention measure. The third output variable was the difference between (1) the perceived chance (0–100%) of contracting COVID-19 if one does not change their lifestyle at all and (2) the perceived chance (0–100%) of contracting COVID-19 if one follows all the public health guidelines for their culture with high fidelity, to precisely measure the subjects’ beliefs in effectiveness of public health measures. 

Schematic of the main trust categories explored for examination of their effects on public health measure compliance (mask wearing and social distancing) and belief in effectiveness of the measures. In this article, we especially focus on examining the separate effects of trust from national public health institutions versus national political leaders (black arrows). 

Following prior work emphasizing the role of trust in national political leaders [18] or trust in government generally [14,75,76] in securing public health compliance, we examined the separate roles of each type of government trust: national public health institutions (shortened to “national institutions”), national political leaders, and local institutions (hospitals and officials). 

For regression analysis, an index was constructed for each government trust category based on the mean of responses to two questions which were statistically and conceptually correlated within that category (Methods) (Appendix A). For trust in national public health institutions, the two questions were about trust in pandemic-related information from national public health institutions, and confidence in the competence of national public health institutions. Similarly, for trust in national political leaders, the two questions were about trust in pandemic-related information from the leaders, and confidence in the competence of the leaders. Supplementary regression was performed, with each of the two national government trust measures as the dependent variables, to verify that the two national government trust categories, political leaders versus institutions, were distinct from each other (Appendix A). We found significant differences in the relationships of two government trust types with demographic and other variables (Appendix A). Last, for local institutional trust, the questions were about trust in hospitals and trust in local officials. The separate contributions of the two individual questions within each category were later examined with structural equation modeling (Appendix A).

#### 3.1.1. Government Trust & Public Health Compliance Behaviors and Beliefs

In the main regression results (Table 1), national institutional trust and local institutional trust were found to have significant positive coefficients for improving compliance of mask wearing and social distancing, as well as for increasing belief in the effectiveness of following these public health measures. In contrast, trust in national political leaders was not predictive of any public health behavior, with weak negative coefficients for compliance and belief in effectiveness of measures (Table 1). These results were robust across all supplementary models explored during model comparison, including a random intercept model (Appendix A) (Appendix A), and separately running individual linear models for each country/territory (Appendix A) (Appendix A). The regression analyses were repeated in both STATA and R for validation. The STATA output containing *p* values and confidence intervals for the Table 1 regression results is provided in Appendix A.

Within the regression analysis (Table 1, Figure 2), independent variables were standardized while dependent variables were preserved in their original units (%, 0–1 normalized scale), thus the regression coefficients report the % change in compliance or belief in effectiveness for every 1 standard deviation (SD) change in each independent variable. On average, each trust measure had an SD~2 units on our 1–10 survey scale (Appendix A). Thus, for example, 2.2 units in increase of national institutional trust (SD) (Appendix A) provided an increase in 4.8% of belief in the effectiveness of public health measures globally (Table 1). For the dependent variables, 6.6 +/− 4.6% (5.1 +/− 2.6%) (SD) of subjects per country/territory were excluded from the mask-wearing (social-distancing) regression for saying they were never in a situation requiring mask wearing (social distancing); 2.6 +/− 1.5% of subjects per country/territory were excluded from the belief in effectiveness of measures regression for saying they were never in a situation requiring masks and social distancing. 

One might wonder whether the effects of these trust measures on compliance behaviors vary across cultures. To examine this cross-cultural variation of the effects of trust in governments, we next broke down the effects of each of the three types of government trust coefficients across individual cultures (Figure 2). For national institutional trust, the vast majority of countries/territories had positive effects on compliance measures, while local institutional trust was somewhat more heterogenous, but still positive on average for improving public health compliance and the perceived effectiveness of these measures (Figure 2). The regression coefficients for trust in political leaders were more variable and inconsistent across geographical regions, but weakly negative on average for all public health behaviors. Additionally, differences in the baseline public health behavior variables across countries/territories are summarized in Appendix A. 

In summary, within our COVID-19 pandemic survey results, trust in national public health institutions was universally the most consistent channel of government-related trust for inducing public health compliance, whereas the trust in national leaders was not significantly predictive of public health compliance. This result agrees with prior results that the public historically places high trust and responsibility on public health officials during a public health crisis [10,11,32,33]. Trust in local public health institutions was also important in our results (Table 1, Figure 2), agreeing with the general framework outlined by early trust researchers about the importance of fostering trust at the local scale for success in national endeavors broadly [8,9,13].

#### 3.1.2. Transparency’s Mediating Impact on Government Trust and Public Health Behavior

The finding that institutional trust towards public health agencies and experts is critical for securing public health compliance leads to the question at the public policy level: What government behaviors can be done to better foster trust between the public and their health-related institutions to promote cooperation in public health measures? Transparency, especially from public health agencies, has long been suspected to be important in improving trust towards public health responses during pandemics, but understanding the nature of the impact of transparency on public trust has been historically difficult in policy research [35,43].

To examine the role of transparency in inducing cooperation in a society, we first tested the relationships among the three types of government trust (national and local institutions, national political leader), transparency, and public health behaviors (public health measure compliance and beliefs in the effectiveness of these measures) in structural equation models (SEM) (Figure 3). The modeling approach of SEM is useful because it can be used to distinguish the direct effect of a given input variable on an output variable and the indirect effect through a mediator variable on the same output variable—a task that cannot be performed easily in regression models [77]. SEM also provides the flexibility to identify the endogenous and the exogenous variables to be tested in the model, thereby allowing researchers to provide additional evidence suggesting possible causal relations rather than mere correlations of the variables of interest.

Our SEM results (Figure 3, Appendix A) show that when transparency enhances government trust, this indirect pathway does increase public health compliance and beliefs in all three categories of government trust. Agreeing with the primary regression results (Table 1), we found that the strongest positive effect from transparency on public health behaviors occurs when transparency fosters national institutional trust in public health institutions. We found a moderate positive effect from transparency on public health behaviors through trust in local institutions. Finally, we found the weakest effect, though positive, from transparency when it increases trust in national political leaders. Additionally, the strong effects of transparency on all types of government trust were confirmed by regression analysis (Appendix A). However, uncoupled to government trust, the direct pathway from transparency to public health behaviors actually had negative weighting hindering public health behaviors or had a neutral effect depending on the model. The inconsistent or negative effects from transparency alone on public health behaviors was also confirmed by regression analysis (Appendix A). The detailed STATA output for all SEM results is provided in Appendix A.

In summary, both linear regression analysis and SEM show that transparency can have an indirect effect on public health behaviors through institutional trust, whereas transparency alone does not increase public health compliance or belief in the effectiveness of these measures. These results may suggest that when transparency is used effectively in policy making and in official communications, it can improve trust in national institutions, and in turn has a strong positive effect on securing public health compliance and belief in their effectiveness.

### 3.2. Secondary Results

Due to the extensive list of trust questions in our survey, we were further able to separately comment on a few recent high-interest trust-related COVID-19 topics in the literature, as provided briefly below. Readers may find these secondary results useful replications or expansions of recent important COVID-19 findings, and these secondary replications helped solidify the quality of our survey data. 

#### 3.2.1. Trust in Science

A prior study has reported that across cultures, higher trust in science improves public health compliance [78]. We replicated this report with a strong positive coefficient from the effect from trust in science for most countries/territories studied (Table 1, Appendix A). Such results in the pandemic context can be complicated to interpret. For example, Japan, Taiwan and South Korea were previously found to have unusually low baseline trust in science among democracies [79]. We also found that these cultures benefited least from trust in science for improving public health compliance (Appendix A). Yet these East Asian cultures have among the best pandemic outcomes overall at the global scale [47]. The relatively low COVID-19 cases and deaths seen in East Asia may instead be partly due to their recent historical institutional experience with related infectious disease epidemics that occurred in these geographical regions [52,80]. 

#### 3.2.2. Trust in Social Media and Traditional News Media

Much work has warned that social media platforms in Western societies rapidly spread misinformation, and that, compounding the misinformation problem, there is potentially worsening quality of traditional news media globally [25,81,82]. These information media-related problems may erode public trust in institutional information sources that encourage public health compliance behaviors [81] (Appendix A). In our results, however, trust in social media and trust in traditional news media were not predictors of public health compliance (Table 1). Higher levels of trust in social media were actually correlated with lower belief in the effectiveness of measures globally (Table 1), aligning with the well-documented prevalence of misinformation on these platforms [25,81]. While there was some geographic variation of effects of social media and traditional new media on the compliance behavior, overall, the effects were neutral and insignificant (Appendix A). 

#### 3.2.3. Trust in Local Communities & General Trust

Representative researchers of trust and cooperation have written extensively on the importance of trust in local communities and general trust for achieving a variety of different large scale cooperative enterprises within societies [2,8,9,13]. We verified this concept in the pandemic context, finding in the regression results (Table 1, Appendix A) and in supplementary structural equation modeling (Appendix A) that higher trust in friends and family (local community trust) was correlated with higher compliance and belief in the effectiveness of public health measures, though Yamagishi’s standard general trust questions [2] were not significant predictors (Table 1). On the other hand, lower trust in strangers and acquaintances was found to correlate with higher public health compliance and increased belief in the effectiveness of measures (Table 1, Appendix A). This last finding agrees with a prior report, where lower social trust was found to correlate with stronger identification of strangers as infection threats [83].

#### 3.2.4. Demographic Trends

Although demographics were all generally weaker and less consistent predictors of public health behaviors than national institutional trust and trust in science (as summarized in Table 1), there were some notable patterns from these variables in predicting public health behaviors. Being female had the greatest positive effects on compliance in public health measures, and more conservative political ideology (the measure was ranked liberal to conservative on 1–10 scale) had the greatest negative effect among demographic variables in our survey (Table 1), agreeing with prior works [23,84]. Older age also had moderate positive effects on compliance behaviors. Last, regarding effects of the demographic variables on transparency, education levels were uncorrelated with transparency question responses (Appendix A), suggesting that subjects of any educational background are able to feel that public health advice has been properly explained to them. National identity, two exploratory independent variables [85] in Table 1, is discussed in more detail in the Appendix A. Future work with more specific demographically targeted surveys could consider further probing these trends. 

Additionally, to demonstrate there were no multicollinearity problems in the regression’s independent variables, a variance inflation factor (VIF) analysis was run, finding a mean VIF value of 1.86 across Table 1′s independent variables (no individual VIF value was greater than 4) (Appendix A). Generally, demographics per country/territory were fairly well distributed by age, gender, income, race, education, etc. according to each country/territory’s own domestic situation. Western countries generally had slightly fewer female than male respondents (44.3 +/− 9.3% female, SD) while Asian countries/territories generally had slightly more female respondents (53.0 +/− 2.0% female, SD) (Appendix A). Overall, there was a moderate general skew towards younger, more educated, less politically conservative, and less religious participants in most countries/territories (Appendix A). This skew was a possible limitation of our survey, though is expected in online-based survey research [18]. Note however that American or Western notions of religiosity and political ideology do not necessarily extend well to Asian cultures, so globally, these scales should be interpreted carefully. See Appendix A for more extensive summaries about survey demographics.

## 4. Discussion

Within the topics covered by our multiscale trust survey, we have demonstrated that national and local institutional trust, rather than trust in national political leaders, are among the most consistent predictors of public health compliance behaviors in a universal way that transcends cultural and geographic variations. This finding cautions against popular narratives that interpersonal trust towards political leaders is central for cooperative behaviors of citizens during a crisis [18,22,23,36]. Instead, our results suggest that the public may generally rely more on expert institutional guidance than their political leaders, at least during a public health crisis. Overall, our fine-grained examinations of trust in society and transparency provide a clear picture of the important socio-psychological factors that influence society-level cooperation during long-term crises within the specific context of the COVID-19 pandemic. 

Future work attempting to place survey results about trust such as ours into broader theoretical frameworks should seek to further untangle the different types of government trust. Each possible government trust sub-category (e.g., institutions generally, staff versus agency heads, elected political leaders, etc.) may be influenced by separate contributions from the core dimensions of interpersonal trust, institutional trust, and political trust [36]. In our context, national political leader trust is expected to be more influenced by the interpersonal and political dimensions, and trust in public health agencies is expected to be more influenced by the institutional trust dimension [28,36,37,38]. Theoretically, the institutional trust dimension is recognized to be more influenced by legal regulations and precedence, civic norms, and social contracts, and is further strengthened by agencies being comprised of staff trained to fulfill the trusted social roles of scientific and medical experts [30,34,36,79,86]. Thus, our finding that national institutional trust is more predictive than trust in national political leaders for securing public health compliance during a pandemic makes conceptual sense, due to the inherent differences in the nature of political leader trust versus institutional trust [34]. Indeed, public health officials and agencies are historically highly trusted and recognized to have the core responsibility, specialized expertise, and training necessary to respond to evolving public health crises, which the public know political leaders generally do not have [11,31,32,33].

While public health institutions played a major role in preventative behavior compliance, as a possible limitation of this work, employers and managers also were important institutional influences for daily public health compliance decisions [87], a topic for which our “Trust in Employers” (Table 1) question only scratched the surface in interrogation. Thus, future work should also consider investigating in more detail the important balance between public health institutional trust and corporate institutional trust across different societies.

For considering actions or policies that can be done to improve institutional trust in societies, it is important to note that the public in modern democracies now expects more transparency and inclusion during government decision making [43]. Policy implementation is more successful when the public’s needs for explanation are met [31,70]. As became clear from our transparency questions in the survey, the public trusts their public health institutions more when the institutions provide convincing rationale behind disaster response guidelines, recognize public feedback, and provide flexible policies that recognize that different populations within a society may have different values and needs (e.g., religious versus secular, rural versus urban, etc.) [43,88]. Admissions of uncertainty in official communications are also crucial and expected in most cultures, as they signal honesty and openness from public health officials [32,33,40,78]. This trend is especially true in novel evolving crises where established playbooks do not exist and the public will not expect complete confidence from their officials [40]. However, optimizing transparency in public health is tricky [35,40,89]. For example, too much uncertainty can cause panic [40,89], but downplaying uncertainty to achieve short-term increases in institutional trust may backfire in the long-term if prior confident public statements are later amended or retracted, or if the public cannot understand why agencies are selectively trusting some scientific experts but not others [35,42,58,90]. Nevertheless, strengthening these institutional trust pathways through transparency policies is very important for securing long-term compliance from the public during prolonged crises. This potential policy mechanism can be partly supported by our findings that institutional trust predicts significantly higher beliefs in the effectiveness of public health measures, which would influence compliance in public health behavior in the long run. 

Additionally, during complex public health crises, the public generally lacks the ability to translate scientific results and expert opinions into effective action without institutional guidance. This issue may be one main reason why trust in specialized institutions is so important for inducing compliance in public health measures. Thus, more effort towards science communication may be necessary to improve scientific trust during crises periods that give boost to public interest and confidence in scientific knowledge [35,91]. Indeed, in our results (Appendix A), trust in science was found to improve both compliance and beliefs in effectiveness of public health measures, consistent with prior work [78]. However, broader problems in the scientific community can also have major impacts on trust towards scientifically oriented government institutions [92]. For example, the on-going replication crisis generally in the biomedical and social sciences (that has been widely reported to the public) [93], paired with the rush-to-publish emergency mindset during the pandemic, has resulted in many COVID-19 articles being retracted [94]. These situations led to some understandable justification for the public to be more skeptical of many public health experts [91]. Because such general trust in sources of scientific information could be important for the management of future crises [92], both governments and scientists may have to spend more effort in nurturing the trust of the public in scientific institutions specifically and the scientific method generally.

## 5. Conclusions

In summary, our international cross-cultural survey revealed that institutional trust is universally essential in encouraging the public to follow COVID-19 prevention measures. In contrast, more interpersonal trust in national political leaders was a poor predictor for securing public health compliance. At the policy-level, transparency was found to indirectly influence good public health behaviors by increasing government trust, with the strongest positive effect on public health seen when transparency increased national institutional trust. Notably, these results were reached by separating out components of government trust into more components than is traditionally done in sociological trust research. We then probed the finer-grained government trust components by looking at both direct correlational links (regression) and indirect correlational links (structural equation modeling) between trust and actions/beliefs. Thus, our research suggests that to more practically embed sociological results in complex real-world contexts, survey research especially may need to inquire about topics with more nuance and detail even at the cost of longer surveys. 

Looking more broadly to the future, prior trust research has warned that fostering institutional trust and building systems of cooperation in society is a slow, long-term process that can easily be set back by institutional missteps [15,26,28,38,39,42]. Our results further emphasize that carefully building and maintaining institutional trust is essential to improving the emergency preparedness of our societies across the globe, and that transparency is vital for building this trust. A limitation of this work however is that we were not able to provide detailed case studies of the local and national institutional public health policies within each country/territory (strategies which often varied significantly even over short time frames during the pandemic). Future work should seek to perform more detailed investigations and comparisons into the institutional strategies that individual countries/territories used during the COVID-19 pandemic, in order to more deeply ground abstract sociological trust results within more detailed policy decisions. 

## Figures and Tables

**Figure 1 behavsci-12-00170-f001:**
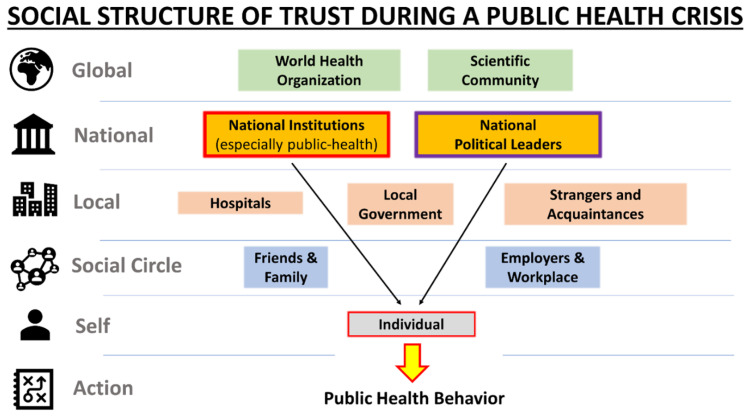
Hierarchical structure of trust during a public health crisis.

**Figure 2 behavsci-12-00170-f002:**
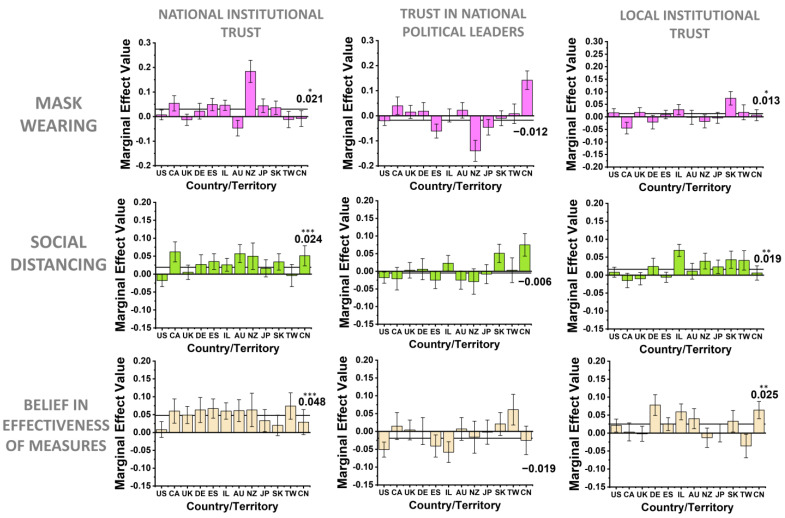
Per country/territory marginal effects for government-related trust components’ influence on public health actions and beliefs. Heterogenous per country/territory marginal effects for the government-related trust categories of national institutional trust in public health agencies, trust in national political leaders, and trust in local institutions (hospitals and officials), for each public health behavior variable: mask wearing, social distancing, and belief in the effectiveness of public health measures. Independent variables only are standardized; dependent variables are kept as percentages (0 to 1 normalized scale). Error bars are standard errors of the mean, and the global coefficient value from Table 1 is marked with the horizontal line on each plot (* *p* < 0.10; ** *p* < 0.05; *** *p* < 0.01). Standard ISO 3166 abbreviations for each country/territory are used.

**Figure 3 behavsci-12-00170-f003:**
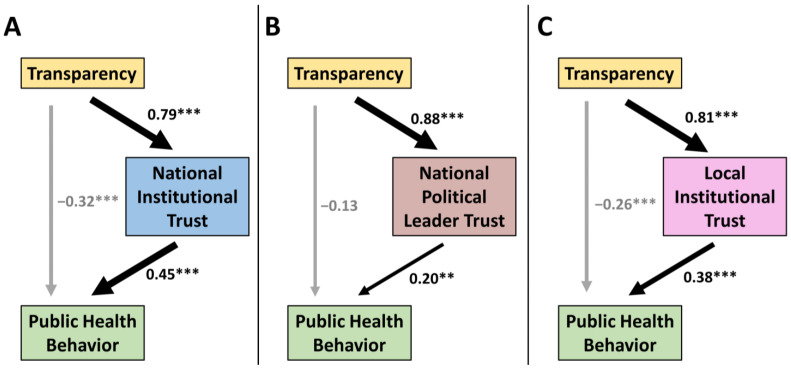
Exploration of causal effects of transparency on different components of government trust. Summary of the three government trust structural equation models tested to explore the causal effects of transparency on public health behavior: (**A**) national institutional trust, (**B**) national political leader trust, and (**C**) local institutional trust. Standardized coefficients and their significance are reported for each pathway (** *p* < 0.01; *** *p* < 0.001). The *p* values in (**A**) and (**C**) are all less than 0.001. In (**B**), the *p* value for the −0.13 coefficient is 0.060, for the 0.20 coefficient is 0.003, and for the 0.88 coefficient is less than 0.001. The RMSEA for each model is (**A**) 0.074, (**B**) 0.064, and (**C**) 0.058 (with all models having an RMSEA upper bound <0.08 in their confidence intervals), and the CFI for each model was (**A**) 0.961, (**B**) 0.971, and (**C**) 0.972. Thus, both RMSEA and CFI each indicate a high-quality fit for all three models.

**Table 1 behavsci-12-00170-t001:** Linear regression summary of the impact of trust variables on public health measure compliance (masks and social distancing) and belief in effectiveness of health measures *.

	(1)	(2)	(3)
VARIABLES	Mask-Wearing	Social-Distancing	Belief in Effectiveness of Measures
Trust in WHO	0.004	0.003	−0.005
	(0.006)	(0.006)	(0.008)
Trust in Science	0.020 **	0.022 **	0.028 ***
	(0.007)	(0.008)	(0.007)
National Institutional Trust	0.021 *	0.024 ***	0.048 ***
	(0.010)	(0.007)	(0.005)
National Political Leader Trust	−0.012	−0.006	−0.019
	(0.011)	(0.007)	(0.011)
Local Institutional Trust	0.013 *	0.019 **	0.025 **
	(0.006)	(0.007)	(0.008)
Trust in Strangers	−0.042 ***	−0.030 ***	−0.066 ***
	(0.005)	(0.005)	(0.008)
Trust in Employers	0.002	0.006	0.003
	(0.008)	(0.006)	(0.007)
Local Community Trust	0.014 **	0.016 ***	0.010 *
	(0.006)	(0.004)	(0.005)
Social Media Trust	0.003	0.002	−0.021 **
	(0.006)	(0.007)	(0.007)
Traditional Media Trust	−0.001	−0.008	0.005
	(0.005)	(0.005)	(0.006)
General Trust (Local)	0.007	0.009	0.008
	(0.006)	(0.005)	(0.008)
General Trust (Global)	−0.007	−0.002	−0.011
	(0.008)	(0.003)	(0.011)
Gender (Female+)	0.025 ***	0.038 ***	0.024 **
	(0.008)	(0.008)	(0.010)
Education Level	0.009 *	0.005	0.008
	(0.005)	(0.005)	(0.005)
Medical Experience	0.006 *	0.003	0.002
	(0.003)	(0.003)	(0.007)
Income	−0.001	−0.010 *	0.005
	(0.008)	(0.005)	(0.004)
Sufficient Safety Net	0.016 **	0.007	0.003
	(0.005)	(0.006)	(0.005)
# of Household Minors	−0.014 ***	0.001	−0.006
	(0.003)	(0.005)	(0.007)
Political Ideology (Conservative+)	−0.012 **	−0.004	−0.022 ***
	(0.005)	(0.005)	(0.006)
Religiosity	−0.009	−0.004	−0.007
	(0.005)	(0.005)	(0.006)
Urbanicity	0.013 **	−0.005	0.011 *
	(0.006)	(0.007)	(0.005)
Experienced Pandemic Financial Hardship	−0.002	0.006	−0.015 **
	(0.005)	(0.006)	(0.006)
Age Group	0.009 *	0.021 ***	−0.007
	(0.004)	(0.003)	(0.005)
National Identity 1	0.021 *	0.005	0.025 ***
	(0.010)	(0.009)	(0.006)
National Identity 2	−0.006	0.003	0.003
	(0.005)	(0.006)	(0.007)
Observations	3113	3155	3240
R-squared	0.182	0.138	0.235

Standard errors in the parentheses are clustered at country/territory level; * *p* < 0.10; ** *p* < 0.05; *** *p* < 0.01; * Primary regression table for the effects of different trust categories, national identity and demographics on the public health behavior variables of mask wearing, social distancing, and belief in the effectiveness of public health measures. Standardized regressors were used.

## Data Availability

Appendix A are publicly available at the Open Science Framework (OSF) directory https://osf.io/sevct/ (accessed on 24 May 2022).

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
