# Peer review of "Trust in Institutions, Not in Political Leaders, Determines Compliance in COVID-19 Prevention Measures within Societies across the Globe"

_behavsci, 2022, doi:10.3390/bs12060170_

Round 1
Reviewer 1 Report
- In the Introduction the authors substantiate the difference between the types of trust, attitudes towards protective measures against Covid-19, and so on. However, the presented study is empirical, which implies the existence of hypotheses, a description of the methodology for testing them, and the interpretation of the results as confirming or refuting the hypotheses put forward. In addition to the general hypothesis, I propose to formulate particular hypotheses corresponding to different stages / methods of research and data processing.
- The criteria for selecting countries participating in the survey are not clear. Indeed, there is a fairly large difference between the countries whose residents took part in the survey. However, the logic behind data collection and country selection is not clear and needs to be spelled out. For example, on what grounds and whether it is correct to compare the results that were obtained regarding the political leader trust in the United States of America, the United Kingdom, Canada, Germany, France, Spain, Australia, New Zealand, Japan, South Korea, Taiwan, and China without geopolitical remarks? I recommend to provide the author's justification for the choice of countries for the survey and assumptions about the similarities and differences between their results.
- Figure 1 (Hierarchical structure of trust during a public health crisis) presented by the authors describes the “social circle” as one of the levels, which includes family and friends. At the same time the sample of people that was interviewed are adults, logically, in addition to family and friends, most people have work environment. Work during the COVID-19 period also turned out to be an important factor in the attitude to the pandemic: someone lost it or went to a remote form, but someone, on the contrary, got more opportunities. These factors, which are also related to the social circle, influenced trust, attitudes to anti-pandemic measures, and etc. If the authors excluded this item from the social environment of people, then what was the reason for their assumption?
Reviewer 2 Report
The research question is topical and brings a novel approach regarding the real impacts of COVID-19 pandemic restrictions on the public institutions and population interaction.
The scope of the research was to identify which specific domains of government and social trust were most crucial for securing public health compliance (frequency of mask-wearing and social distancing) and understanding the reasons for following the health measures (belief ineffectiveness of public health measures).
The Title –has a different format – but expresses quite well the focus of the paper.
Abstract: Even if it is brief, the abstract has a standard formulation and presents concisely the purpose and the results of the research. A brief note about the methodology should be included in the abstract. There is too long a description of the topic.
KEYWORDS – are accordingly.
The Introduction: is complying with its scope and insists on convincing about the importance of the study in the population security domain. More references should be used with papers that discuss the COVID-19 effects and implementation strategies.
“COVID‐19 induced emergent knowledge strategies”, Knowledge and Process Management, 28 (1), 11-17.
The Introduction – includes also a “Literature review”.
Literature Revies – Context
The literature review should be enriched with references for supporting the same concept or idea/ or theory. As we mentioned, it is included in the Introduction section.
Research design
This section is atypically structured and presented. Each component theory is argued and also each assumption is supported.
Hypotheses are differently formulated rather than as relations between variables.
Maybe for a such large audience, the authors should clearly formulate, using the phrases “hypothesis”, “objectives” or “relations”.
The research methodology is briefly presented, synthetic, and clearly enough in order to understand the main purpose of the proposed model.
Results and discussion
The Results section is highly well presented on paragraphs related to each investigated component and hypothesis of the SEM model.
For each component and factor are presented and discussed the values and significance for the analysis are according to the SEM protocol.
The presentation of the results is very well detailed and argued.
Impact and Limitations
Conclusions are too brief and don’t formulate an overview of the paper.
Conclusions should clearly point out the theoretical contributions, the practical implications, the limits, and the future of the research.
The authors should achieve to highlight the importance and the practical relevance of the investigation.
References
References are rather centered on a certain area of the globe and they should reflect wider and more diverse views. Could be improved.
Reviewer 3 Report
The study aimed to better understand which entities the public give the most attention to during crises such as COVID-19. It is an unique and essential study. However, there are several points to improve.
- The title should be more specific. The paper deals with compliance (adherence) to preventive measures such as masks and social distancing.
- In Intruduction, The concrete definition of "Trust" should be given at the beginning of it. Also, the authors described that Trust is a powerful predictor of human behavior. How is it so powerful?
- The authors should consider the international differences in health policies among the study countries and territories because healthcare systems and people's attitudes toward it affect "Trust". However, there is little description of international comparisons from Intruduction to Discussion. I think that there is a huge difference among "western countries".
- In the last paragraph of Discussion, there seems to be a leap of logic. This paper is all about COVID-19, but not about a various kinds of disasters such as global warming.
Round 2
Reviewer 3 Report
The paper is improved along with the comments, and seems to be suitable for publication.
Author Response
We thank the reviewer for their helpful suggestions in improving our manuscript in round 1, and acknowledge no additional suggestions were provided in round 2.